# $f$-GAN: Training Generative Neural Samplers using Variational Divergence Minimization

**Sebastian Nowozin, Botond Cseke, Ryota Tomioka**
Machine Intelligence and Perception Group
Microsoft Research
{Sebastian.Nowozin, Botond.Cseke, ryoto}@microsoft.com

## Abstract

Generative neural samplers are probabilistic models that implement sampling using feedforward neural networks: they take a random input vector and produce a sample from a probability distribution defined by the network weights. These models are expressive and allow efficient computation of samples and derivatives, but cannot be used for computing likelihoods or for marginalization. The *generative-adversarial* training method allows to train such models through the use of an auxiliary discriminative neural network. We show that the generative-adversarial approach is a special case of an existing more general variational divergence estimation approach. We show that any $f$-divergence can be used for training generative neural samplers. We discuss the benefits of various choices of divergence functions on training complexity and the quality of the obtained generative models.

## 1 Introduction

Probabilistic generative models describe a probability distribution over a given domain $\mathcal{X}$, for example a distribution over natural language sentences, natural images, or recorded waveforms.

Given a generative model $Q$ from a class $\mathcal{Q}$ of possible models we are generally interested in performing one or multiple of the following operations:

- *Sampling.* Produce a sample from $Q$. By inspecting samples or calculating a function on a set of samples we can obtain important insight into the distribution or solve decision problems.
- *Estimation.* Given a set of iid samples $\{x_1, x_2, \ldots, x_n\}$ from an unknown true distribution $P$, find $Q \in \mathcal{Q}$ that best describes the true distribution.
- *Point-wise likelihood evaluation.* Given a sample $x$, evaluate the likelihood $Q(x)$.

*Generative-adversarial networks* (GAN) in the form proposed by [10] are an expressive class of generative models that allow exact sampling and approximate estimation. The model used in GAN is simply a feedforward neural network which receives as input a vector of random numbers, sampled, for example, from a uniform distribution. This random input is passed through each layer in the network and the final layer produces the desired output, for example, an image. Clearly, sampling from a GAN model is efficient because only one forward pass through the network is needed to produce one exact sample.

Such probabilistic feedforward neural network models were first considered in [22] and [3], here we call these models **generative neural samplers**. GAN is also of this type, as is the decoder model of a variational autoencoder [18].

In the original GAN paper the authors show that it is possible to estimate neural samplers by approximate minimization of the symmetric *Jensen-Shannon divergence*,

$$D_{\text{JS}}(P\|Q) = \tfrac{1}{2}D_{\text{KL}}(P\|\tfrac{1}{2}(P+Q)) + \tfrac{1}{2}D_{\text{KL}}(Q\|\tfrac{1}{2}(P+Q)), \qquad (1)$$

where $D_{KL}$ denotes the Kullback-Leibler divergence. The key technique used in the GAN training is that of introducing a second "*discriminator*" neural networks which is optimized simultaneously.

Because $D_{\mathrm{JS}}(P\|Q)$ is a proper divergence measure between distributions this implies that the true distribution $P$ can be approximated well in case there are sufficient training samples and the model class $\mathcal{Q}$ is rich enough to represent $P$.

In this work we show that the principle of GANs is more general and we can extend the variational divergence estimation framework proposed by Nguyen et al. [25] to recover the GAN training objective and generalize it to arbitrary $f$-divergences.

More concretely, we make the following contributions over the state-of-the-art:

- We derive the GAN training objectives for all $f$-divergences and provide as example additional divergence functions, including the Kullback-Leibler and Pearson divergences.
- We simplify the saddle-point optimization procedure of Goodfellow et al. [10] and provide a theoretical justification.
- We provide experimental insight into which divergence function is suitable for estimating generative neural samplers for natural images.

## 2 Method

We first review the divergence estimation framework of Nguyen et al. [25] which is based on $f$-divergences. We then extend this framework from divergence estimation to model estimation.

### 2.1 The f-divergence Family

Statistical divergences such as the well-known *Kullback-Leibler divergence* measure the difference between two given probability distributions. A large class of different divergences are the so called $f$-divergences [5, 21], also known as the Ali-Silvey distances [1]. Given two distributions $P$ and $Q$ that possess, respectively, an absolutely continuous density function $p$ and $q$ with respect to a base measure $\mathrm{d}x$ defined on the domain $\mathcal{X}$, we define the $f$-divergence,

$$D_f(P\|Q) = \int_{\mathcal{X}} q(x) f\left(\frac{p(x)}{q(x)}\right)\,\mathrm{d}x, \tag{2}$$

where the *generator function* $f : \mathbb{R}_+ \to \mathbb{R}$ is a convex, lower-semicontinuous function satisfying $f(1) = 0$. Different choices of $f$ recover popular divergences as special cases in (2). We illustrate common choices in Table 1. See supplementary material for more divergences and plots.

### 2.2 Variational Estimation of $f$-divergences

Nguyen et al. [25] derive a general variational method to estimate $f$-divergences given only samples from $P$ and $Q$. An equivalent result has also been derived by Reid and Williamson [28]. We will extend these results from merely estimating a divergence for a fixed model to estimating model parameters. We call this new method *variational divergence minimization* (VDM) and show that generative-adversarial training is a special case of our VDM framework.

For completeness, we first provide a self-contained derivation of Nguyen et al's divergence estimation procedure. Every convex, lower-semicontinuous function $f$ has a *convex conjugate* function $f^*$, also known as *Fenchel conjugate* [15]. This function is defined as

$$f^*(t) = \sup_{u \in \mathrm{dom}_f} \{ut - f(u)\}. \tag{3}$$

The function $f^*$ is again convex and lower-semicontinuous and the pair $(f, f^*)$ is dual to another in the sense that $f^{**} = f$. Therefore, we can also represent $f$ as $f(u) = \sup_{t \in \mathrm{dom}_{f^*}} \{tu - f^*(t)\}$. Nguyen et al. leverage the above variational representation of $f$ in the definition of the $f$-divergence to obtain a lower bound on the divergence,

$$
\begin{aligned}
D_f(P\|Q) &= \int_{\mathcal{X}} q(x) \sup_{t \in \mathrm{dom}_{f^*}} \left\{ t \tfrac{p(x)}{q(x)} - f^*(t) \right\}\,\mathrm{d}x \\
&\geq \sup_{T \in \mathcal{T}} \left( \int_{\mathcal{X}} p(x)\,T(x)\,\mathrm{d}x - \int_{\mathcal{X}} q(x)\,f^*(T(x))\,\mathrm{d}x \right) \\
&= \sup_{T \in \mathcal{T}} \left( \mathbb{E}_{x \sim P}\left[T(x)\right] - \mathbb{E}_{x \sim Q}\left[f^*(T(x))\right] \right),
\end{aligned}
\tag{4}
$$

| Name | $D_f(P\|Q)$ | Generator $f(u)$ | $T^*(x)$ |
|---|---|---|---|
| Kullback-Leibler | $\int p(x) \log \frac{p(x)}{q(x)} \, \mathrm{d}x$ | $u \log u$ | $1 + \log \frac{p(x)}{q(x)}$ |
| Reverse KL | $\int q(x) \log \frac{q(x)}{p(x)} \, \mathrm{d}x$ | $-\log u$ | $-\frac{q(x)}{p(x)}$ |
| Pearson $\chi^2$ | $\int \frac{(q(x)-p(x))^2}{p(x)} \, \mathrm{d}x$ | $(u-1)^2$ | $2(\frac{p(x)}{q(x)} - 1)$ |
| Squared Hellinger | $\int \left(\sqrt{p(x)} - \sqrt{q(x)}\right)^2 \, \mathrm{d}x$ | $(\sqrt{u} - 1)^2$ | $(\sqrt{\frac{p(x)}{q(x)}} - 1) \cdot \sqrt{\frac{q(x)}{p(x)}}$ |
| Jensen-Shannon | $\frac{1}{2} \int p(x) \log \frac{2p(x)}{p(x)+q(x)} + q(x) \log \frac{2q(x)}{p(x)+q(x)} \, \mathrm{d}x$ | $-(u+1) \log \frac{1+u}{2} + u \log u$ | $\log \frac{2p(x)}{p(x)+q(x)}$ |
| GAN | $\int p(x) \log \frac{2p(x)}{p(x)+q(x)} + q(x) \log \frac{2q(x)}{p(x)+q(x)} \, \mathrm{d}x - \log(4)$ | $u \log u - (u+1) \log(u+1)$ | $\log \frac{p(x)}{p(x)+q(x)}$ |

**Table 1:** List of $f$-divergences $D_f(P\|Q)$ together with generator functions. Part of the list of divergences and their generators is based on [26]. For all divergences we have $f : \mathrm{dom}_f \to \mathbb{R} \cup \{+\infty\}$, where $f$ is convex and lower-semicontinuous. Also we have $f(1) = 0$ which ensures that $D_f(P\|P) = 0$ for any distribution $P$. As shown by [10] GAN is related to the Jensen-Shannon divergence through $D_{\mathrm{GAN}} = 2D_{\mathrm{JS}} - \log(4)$.

where $\mathcal{T}$ is an arbitrary class of functions $T : \mathcal{X} \to \mathbb{R}$. The above derivation yields a lower bound because the class of functions $\mathcal{T}$ may contain only a subset of all possible functions. By taking the variation of the lower bound in (4) w.r.t. $T$, we find that under mild conditions on $f$ [25], the bound is tight for

$$T^*(x) = f'\left(\frac{p(x)}{q(x)}\right),\tag{5}$$

where $f'$ denotes the first order derivative of $f$. This condition can serve as a guiding principle for choosing $f$ and designing the class of functions $\mathcal{T}$. For example, the popular reverse Kullback-Leibler divergence corresponds to $f(u) = -\log(u)$ resulting in $T^*(x) = -q(x)/p(x)$, see Table 1.

We list common $f$-divergences in Table 1 and provide their Fenchel conjugates $f^*$ and the domains $\mathrm{dom}_{f^*}$ in Table 2. We provide plots of the generator functions and their conjugates in the supplementary materials.

## 2.3 Variational Divergence Minimization (VDM)

We now use the variational lower bound (4) on the $f$-divergence $D_f(P\|Q)$ in order to estimate a generative model $Q$ given a true distribution $P$.

To this end, we follow the generative-adversarial approach [10] and use two neural networks, $Q$ and $T$. $Q$ is our generative model, taking as input a random vector and outputting a sample of interest. We parametrize $Q$ through a vector $\theta$ and write $Q_\theta$. $T$ is our variational function, taking as input a sample and returning a scalar. We parametrize $T$ using a vector $\omega$ and write $T_\omega$.

We can train a generative model $Q_\theta$ by finding a saddle-point of the following $f$-GAN objective function, where we minimize with respect to $\theta$ and maximize with respect to $\omega$,

$$F(\theta, \omega) = \mathbb{E}_{x \sim P}\left[T_\omega(x)\right] - \mathbb{E}_{x \sim Q_\theta}\left[f^*(T_\omega(x))\right].\tag{6}$$

To optimize (6) on a given finite training data set, we approximate the expectations using minibatch samples. To approximate $\mathbb{E}_{x \sim P}[\cdot]$ we sample $B$ instances without replacement from the training set. To approximate $\mathbb{E}_{x \sim Q_\theta}[\cdot]$ we sample $B$ instances from the current generative model $Q_\theta$.

## 2.4 Representation for the Variational Function

To apply the variational objective (6) for different $f$-divergences, we need to respect the domain $\mathrm{dom}_{f^*}$ of the conjugate functions $f^*$. To this end, we assume that variational function $T_\omega$ is represented in the form $T_\omega(x) = g_f(V_\omega(x))$ and rewrite the saddle objective (6) as follows:

$$F(\theta, \omega) = \mathbb{E}_{x \sim P}\left[g_f(V_\omega(x))\right] + \mathbb{E}_{x \sim Q_\theta}\left[-f^*(g_f(V_\omega(x)))\right],\tag{7}$$

where $V_\omega : \mathcal{X} \to \mathbb{R}$ without any range constraints on the output, and $g_f : \mathbb{R} \to \mathrm{dom}_{f^*}$ is an *output activation function* specific to the $f$-divergence used. In Table 2 we propose suitable output activation functions for the various conjugate functions $f^*$ and their domains.[1] Although the choice of $g_f$ is somewhat arbitrary, we choose all of them to be monotone increasing functions so that a large output

| Name | Output activation $g_f$ | $\mathrm{dom}_{f^*}$ | Conjugate $f^*(t)$ | $f'(1)$ |
|---|---|---|---|---|
| Kullback-Leibler (KL) | $v$ | $\mathbb{R}$ | $\exp(t-1)$ | $1$ |
| Reverse KL | $-\exp(-v)$ | $\mathbb{R}_-$ | $-1-\log(-t)$ | $-1$ |
| Pearson $\chi^2$ | $v$ | $\mathbb{R}$ | $\frac{1}{4}t^2+t$ | $0$ |
| Squared Hellinger | $1-\exp(-v)$ | $t<1$ | $\frac{t}{1-t}$ | $0$ |
| Jensen-Shannon | $\log(2)-\log(1+\exp(-v))$ | $t<\log(2)$ | $-\log(2-\exp(t))$ | $0$ |
| GAN | $-\log(1+\exp(-v))$ | $\mathbb{R}_-$ | $-\log(1-\exp(t))$ | $-\log(2)$ |

**Table 2:** Recommended final layer activation functions and critical variational function level defined by $f'(1)$. The critical value $f'(1)$ can be interpreted as a classification threshold applied to $T(x)$ to distinguish between true and generated samples.

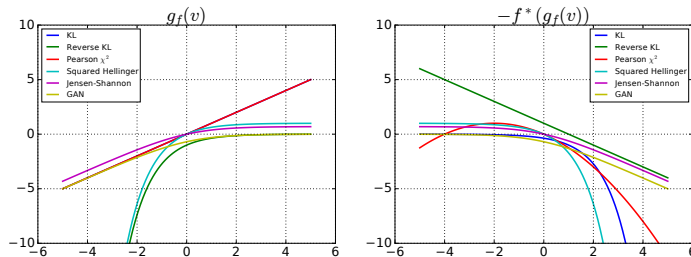

**Figure 1:** The two terms in the saddle objective (7) are plotted as a function of the variational function $V_\omega(x)$.

$V_\omega(x)$ corresponds to the belief of the variational function that the sample $x$ comes from the data distribution $P$ as in the GAN case; see Figure 1. It is also instructive to look at the second term $-f^*(g_f(v))$ in the saddle objective (7). This term is typically (except for the Pearson $\chi^2$ divergence) a decreasing function of the output $V_\omega(x)$ favoring variational functions that output negative numbers for samples from the generator.

We can see the GAN objective,

$$F(\theta,\omega) = \mathbb{E}_{x\sim P}\left[\log D_\omega(x)\right] + \mathbb{E}_{x\sim Q_\theta}\left[\log(1-D_\omega(x))\right], \qquad (8)$$

as a special instance of (7) by identifying each terms in the expectations of (7) and (8). In particular, choosing the last nonlinearity in the discriminator as the sigmoid $D_\omega(x) = 1/(1+e^{-V_\omega(x)})$, corresponds to output activation function is $g_f(v) = -\log(1+e^{-v})$; see Table 2.

## 2.5 Example: Univariate Mixture of Gaussians

To demonstrate the properties of the different $f$-divergences and to validate the variational divergence estimation framework we perform an experiment similar to the one of [24].

**Setup.** We approximate a mixture of Gaussians by learning a Gaussian distribution. We represent our model $Q_\theta$ using a linear function which receives a random $z \sim \mathcal{N}(0,1)$ and outputs $G_\theta(z) = \mu + \sigma z$, where $\theta = (\mu, \sigma)$ are the two scalar parameters to be learned. For the variational function $T_\omega$ we use a neural network with two hidden layers having 64 units each and tanh activations. We optimize the objective $F(\omega, \theta)$ by using the single-step gradient method presented in Section 3. In each step we sample batches of size 1024 from $p(x)$ and $p(z)$ and we use a step-size of $\eta = 0.01$ for updating both $\omega$ and $\theta$. We compare the results to the best fit provided by the exact optimization of $D_f(P\|Q_\theta)$ w.r.t. $\theta$, which is feasible in this case by solving the required integrals in (2) numerically. We use $(\hat{\omega}, \hat{\theta})$ (learned) and $\theta^*$ (best fit) to distinguish the parameters sets used in these two approaches.

**Results.** The left side of Table 3 shows the optimal divergence and objective values $D_f(P\|Q_{\theta^*})$ and $F(\hat{\omega}, \hat{\theta})$ as well as the corresponding (optimal) means and standard deviations. Note that the results are in line with the lower bound property, having $D_f(P\|Q_{\theta^*}) \geq F(\hat{\omega}, \hat{\theta})$. There is a good correspondence between the gap in objectives and the difference between the fitted means and standard deviations. The right side of Table 3 shows the results of the following experiment: (1) we train $T_\omega$ and $Q_\theta$ using a particular divergence, then (2) we estimate the divergence and re-train $T_\omega$ while keeping $Q_\theta$ fixed. As expected, $Q_\theta$ performs best on the divergence it was trained with. We present further details and plots of the fitted Gaussians and variational functions in the supplementary materials.

| | KL | KL-rev | JS | Jeffrey | Pearson |
|---|---|---|---|---|---|
| $D_f(P\|\|Q_{\theta*})$ | 0.2831 | 0.2480 | 0.1280 | 0.5705 | 0.6457 |
| $F(\hat\omega,\hat\theta)$ | 0.2801 | 0.2415 | 0.1226 | 0.5151 | 0.6379 |
| $\mu^*$ | 1.0100 | 1.5782 | 1.3070 | 1.3218 | 0.5737 |
| $\hat\mu$ | 1.0335 | 1.5624 | 1.2854 | 1.2295 | 0.6157 |
| $\sigma^*$ | 1.8308 | 1.6319 | 1.7542 | 1.7034 | 1.9274 |
| $\hat\sigma$ | 1.8236 | 1.6403 | 1.7659 | 1.8087 | 1.9031 |

| train \ test | KL | KL-rev | JS | Jeffrey | Pearson |
|---|---|---|---|---|---|
| KL | **0.2808** | 0.3423 | 0.1314 | 0.5447 | 0.7345 |
| KL-rev | 0.3518 | **0.2414** | 0.1228 | 0.5794 | 1.3974 |
| JS | 0.2871 | 0.2760 | **0.1210** | 0.5260 | 0.92160 |
| Jeffrey | 0.2869 | 0.2975 | 0.1247 | **0.5236** | 0.8849 |
| Pearson | 0.2970 | 0.5466 | 0.1665 | 0.7085 | **0.648** |

**Table 3:** Gaussian approximation of a mixture of Gaussians. Left: optimal objectives, and the learned mean and the standard deviation: $\hat\theta = (\hat\mu, \hat\sigma)$ (learned) and $\theta^* = (\mu^*, \sigma^*)$ (best fit). Right: objective values to the true distribution for each trained model. For each divergence, the lowest objective function value is achieved by the model that was trained for this divergence.

In summary, our results demonstrate that when the generative model is misspecified, the divergence function used for estimation has a strong influence on which model is learned.

# 3 Algorithms for Variational Divergence Minimization (VDM)

We now discuss numerical methods to find saddle points of the objective (6). To this end, we distinguish two methods; first, the alternating method originally proposed by Goodfellow et al. [10], and second, a more direct single-step optimization procedure.

In our variational framework, the alternating gradient method can be described as a double-loop method; the internal loop tightens the lower bound on the divergence, whereas the outer loop improves the generator model. While the motivation for this method is plausible, in practice a popular choice is taking a single step in the inner loop, requiring two backpropagation passes for one outer iteration. Goodfellow et al. [10] provide a local convergence guarantee.

## 3.1 Single-Step Gradient Method

Motivated by the success of the alternating gradient method with a single inner step, we propose an even simpler algorithm shown in Algorithm 1. The algorithm differs from the original one in that there is no inner loop and the gradients with respect to $\omega$ and $\theta$ are computed in a single back-propagation.

---

**Algorithm 1** Single-Step Gradient Method

1: **function** SINGLESTEPGRADIENTITERATION($P, \theta^t, \omega^t, B, \eta$)
2:     Sample $X_P = \{x_1, \ldots, x_B\}$ and $X_Q = \{x'_1, \ldots, x'_B\}$, from $P$ and $Q_{\theta^t}$, respectively.
3:     Update: $\omega^{t+1} = \omega^t + \eta \, \nabla_\omega F(\theta^t, \omega^t)$.
4:     Update: $\theta^{t+1} = \theta^t - \eta \, \nabla_\theta F(\theta^t, \omega^t)$.
5: **end function**

---

**Analysis.** Here we show that Algorithm 1 geometrically converges to a saddle point $(\theta^*, \omega^*)$ if there is a neighborhood around the saddle point in which $F$ is strongly convex in $\theta$ and strongly concave in $\omega$. These assumptions are similar to those made in [10]. Formally, we assume:

$$\nabla_\theta F(\theta^*, \omega^*) = 0, \quad \nabla_\omega F(\theta^*, \omega^*) = 0, \quad \nabla_\theta^2 F(\theta, \omega) \succeq \delta I, \quad \nabla_\omega^2 F(\theta, \omega) \preceq -\delta I, \quad (9)$$

for $(\theta, \omega)$ in the neighborhood of $(\theta^*, \omega^*)$. Note that although there could be many saddle points that arise from the structure of deep networks [6], they would not qualify as the solution of our variational framework under these assumptions.

For convenience, let's define $\pi^t = (\theta^t, \omega^t)$. Now the convergence of Algorithm 1 can be stated as follows (the proof is given in the supplementary material):

**Theorem 1.** *Suppose that there is a saddle point $\pi^* = (\theta^*, \omega^*)$ with a neighborhood that satisfies conditions* (9). *Moreover, we define $J(\pi) = \frac{1}{2}\|\nabla F(\pi)\|_2^2$ and assume that in the above neighborhood, $F$ is sufficiently smooth so that there is a constant $L > 0$ such that $\|\nabla J(\pi') - \nabla J(\pi)\|_2 \le L\|\pi' - \pi\|_2$ for any $\pi, \pi'$ in the neighborhood of $\pi^*$. Then using the step-size $\eta = \delta/L$ in Algorithm 1, we have*

$$J(\pi^t) \le \left(1 - \frac{\delta^2}{L}\right)^t J(\pi^0).$$

*That is, the squared norm of the gradient $\nabla F(\pi)$ decreases geometrically.*

## 3.2 Practical Considerations

Here we discuss principled extensions of the heuristic proposed in [10] and real/fake statistics discussed by Larsen and Sønderby[2]. Furthermore we discuss practical advice that slightly deviate from the principled viewpoint.

Goodfellow et al. [10] noticed that training GAN can be significantly sped up by maximizing $\mathbb{E}_{x \sim Q_\theta}\left[\log D_\omega(x)\right]$ instead of minimizing $\mathbb{E}_{x \sim Q_\theta}\left[\log\left(1 - D_\omega(x)\right)\right]$ for updating the generator. In the more general $f$-GAN Algorithm (1) this means that we replace line 4 with the update

$$\theta^{t+1} = \theta^t + \eta \, \nabla_\theta \mathbb{E}_{x \sim Q_{\theta^t}}[g_f(V_{\omega^t}(x))], \tag{10}$$

thereby maximizing the variational function output on the generated samples. We can show that this transformation preserves the stationary point as follows (which is a generalization of the argument in [10]): note that the only difference between the original direction (line 4) and (10) is the scalar factor $f^{*\prime}(T_\omega(x))$, which is the derivative of the conjugate function $f^*$. Since $f^{*\prime}$ is the inverse of $f'$ (see Cor. 1.4.4, Chapter E, [15]), if $T = T^*$, using (5), we can see that this factor would be the density ratio $p(x)/q(x)$, which would be one at the stationary point. We found this transformation useful also for other divergences. We found *Adam* [17] and gradient clipping to be useful especially in the large scale experiment on the LSUN dataset.

The original implementation [10] of GANs[3] and also Larsen and Sønderby monitor certain *real* and *fake* statistics, which are defined as the true positive and true negative rates of the variational function viewing it as a binary classifier. Since our output activation $g_f$ are all monotone, we can derive similar statistics for any $f$-divergence by only changing the decision threshold. Due to the link between the density ratio and the variational function (5), the threshold lies at $f'(1)$ (see Table 2). That is, we can interpret the output of the variational function as classifying the input $x$ as a true sample if the variational function $T_\omega(x)$ is larger than $f'(1)$, and classifying it as a generator sample otherwise.

# 4 Experiments

We now train generative neural samplers based on VDM on the MNIST and LSUN datasets.

**MNIST Digits.** We use the MNIST training data set (60,000 samples, 28-by-28 pixel images) to train the generator and variational function model proposed in [10] for various $f$-divergences. With $z \sim \text{Uniform}_{100}(-1, 1)$ as input, the generator model has two linear layers each followed by batch normalization and ReLU activation and a final linear layer followed by the sigmoid function. The variational function $V_\omega(x)$ has three linear layers with exponential linear unit [4] in between. The final activation is specific to each divergence and listed in Table 2. As in [27] we use Adam with a learning rate of $\alpha = 0.0002$ and update weight $\beta = 0.5$. We use a batchsize of 4096, sampled from the training set without replacement, and train each model for one hour. We also compare against variational autoencoders [18] with 20 latent dimensions.

*Results and Discussion.* We evaluate the performance using the kernel density estimation (Parzen window) approach used in [10]. To this end, we sample 16k images from the model and estimate a Parzen window estimator using an isotropic Gaussian kernel bandwidth using three fold cross validation. The final density model is used to evaluate the average log-likelihood on the MNIST test set (10k samples). We show the results in Table 4, and some samples from our models in Figure 2.

The use of the KDE approach to log-likelihood estimation has known deficiencies [33]. In particular, for the dimensionality used in MNIST ($d = 784$) the number of model samples required to obtain accurate log-likelihood estimates is infeasibly large. We found a large variability (up to 50 nats) between multiple repetitions. As such the results are not entirely conclusive. We also trained the same KDE estimator on the MNIST training set, achieving a significantly higher holdout likelihood. However, it is reassuring to see that the model trained for the Kullback-Leibler divergence indeed achieves a high holdout likelihood compared to the GAN model.

| Training divergence | KDE $\langle LL \rangle$ (nats) | $\pm$ SEM |
|---|---:|---:|
| Kullback-Leibler | 416 | 5.62 |
| Reverse Kullback-Leibler | 319 | 8.36 |
| Pearson $\chi^2$ | 429 | 5.53 |
| Neyman $\chi^2$ | 300 | 8.33 |
| Squared Hellinger | -708 | 18.1 |
| Jeffrey | -2101 | 29.9 |
| Jensen-Shannon | 367 | 8.19 |
| GAN | 305 | 8.97 |
| Variational Autoencoder [18] | 445 | 5.36 |
| KDE MNIST train (60k) | 502 | 5.99 |

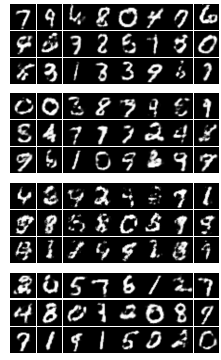

**Table 4:** Kernel Density Estimation evaluation on the MNIST test data set. Each KDE model is build from 16,384 samples from the learned generative model. We report the mean log-likelihood on the MNIST test set ($n = 10,000$) and the standard error of the mean. The KDE MNIST result is using 60,000 MNIST training images to fit a single KDE model.

**Figure 2:** MNIST model samples trained using KL, reverse KL, Hellinger, Jensen from top to bottom.

**LSUN Natural Images.** Through the DCGAN work [27] the generative-adversarial approach has shown real promise in generating natural looking images. Here we use the same architecture as as in [27] and replace the GAN objective with our more general $f$-GAN objective.

We use the large scale LSUN database [35] of natural images of different categories. To illustrate the different behaviors of different divergences we train the same model on the *classroom* category of images, containing 168,103 images of classroom environments, rescaled and center-cropped to 96-by-96 pixels.

*Setup.* We use the generator architecture and training settings proposed in DCGAN [27]. The model receives $z \in \text{Uniform}_{d_{\text{rand}}}(-1, 1)$ and feeds it through one linear layer and three deconvolution layers with batch normalization and ReLU activation in between. The variational function is the same as the discriminator architecture in [27] and follows the structure of a convolutional neural network with batch normalization, exponential linear units [4] and one final linear layer.

*Results.* Figure 3 shows 16 random samples from neural samplers trained using GAN, KL, and squared Hellinger divergences. All three divergences produce equally realistic samples. Note that the difference in the learned distribution $Q_\theta$ arise only when the generator model is not rich enough.

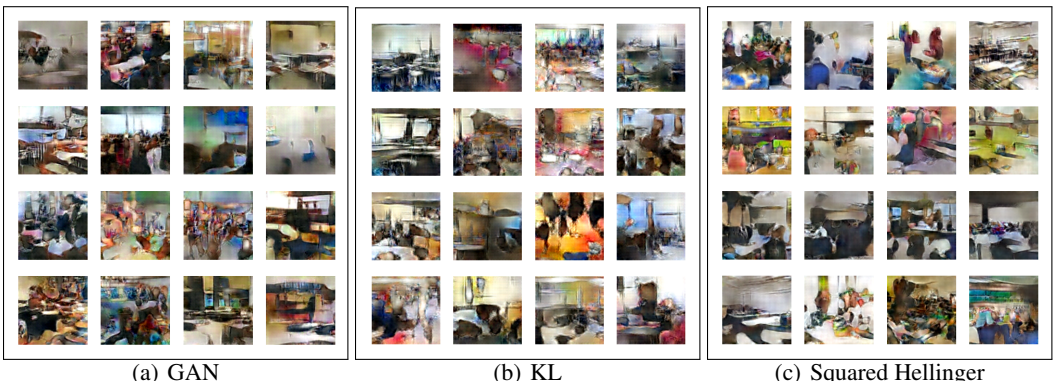

|  (a) GAN  |  (b) KL  |  (c) Squared Hellinger  |

**Figure 3:** Samples from three different divergences.

# 5   Related Work

We now discuss how our approach relates to existing work. Building generative models of real world distributions is a fundamental goal of machine learning and much related work exists. We only discuss work that applies to neural network models.

*Mixture density networks* [2] are neural networks which directly regress the parameters of a finite parametric mixture model. When combined with a recurrent neural network this yields impressive generative models of handwritten text [12].

*NADE* [19] and *RNADE* [34] perform a factorization of the output using a predefined and somewhat arbitrary ordering of output dimensions. The resulting model samples one variable at a time conditioning on the entire history of past variables. These models provide tractable likelihood evaluations and compelling results but it is unclear how to select the factorization order in many applications .

*Diffusion probabilistic models* [31] define a target distribution as a result of a learned diffusion process which starts at a trivial known distribution. The learned model provides exact samples and approximate log-likelihood evaluations.

*Noise contrastive estimation* (NCE) [14] is a method that estimates the parameters of unnormalized probabilistic models by performing non-linear logistic regression to discriminate the data from artificially generated noise. NCE can be viewed as a special case of GAN where the discriminator is constrained to a specific form that depends on the model (logistic regression classifier) and the generator (kept fixed) is providing the artificially generated noise (see supplementary material).

The generative neural sampler models of [22] and [3] did not provide satisfactory learning methods; [22] used importance sampling and [3] expectation maximization. The main difference to GAN and to our work really is in the learning objective, which is effective and computationally inexpensive.

*Variational auto-encoders* (VAE) [18, 29] are pairs of probabilistic encoder and decoder models which map a sample to a latent representation and back, trained using a variational Bayesian learning objective. The advantage of VAEs is in the encoder model which allows efficient inference from observation to latent representation and overall they are a compelling alternative to $f$-GANs and recent work has studied combinations of the two approaches [23]

As an alternative to the GAN training objective the work [20] and independently [7] considered the use of the *kernel maximum mean discrepancy* (MMD) [13, 9] as a training objective for probabilistic models. This objective is simpler to train compared to GAN models because there is no explicitly represented variational function. However, it requires the choice of a kernel function and the reported results so far seem slightly inferior compared to GAN. MMD is a particular instance of a larger class of probability metrics [32] which all take the form $D(P, Q) = \sup_{T \in \mathcal{T}} |\mathbb{E}_{x \sim P}[T(x)] - \mathbb{E}_{x \sim Q}[T(x)]|$, where the function class $\mathcal{T}$ is chosen in a manner specific to the divergence. Beyond MMD other popular metrics of this form are the total variation metric (also an $f$-divergence), the Wasserstein distance, and the Kolmogorov distance.

A previous attempt to enable minimization of the KL-divergence in deep generative models is due to Goodfellow et al. [11], where an approximation to the gradient of the KL divergence is derived.

In [16] another generalization of the GAN objective is proposed by using an *alternative Jensen-Shannon divergence* that interpolates between the KL and the reverse KL divergence and has Jensen-Shannon as its mid-point. We discuss this work in more detail in the supplementary materials.

## 6   Discussion

Generative neural samplers offer a powerful way to represent complex distributions without limiting factorizing assumptions. However, while the purely generative neural samplers as used in this paper are interesting their use is limited because after training they cannot be conditioned on observed data and thus are unable to provide inferences.

We believe that in the future the true benefits of neural samplers for representing uncertainty will be found in discriminative models and our presented methods extend readily to this case by providing additional inputs to both the generator and variational function as in the conditional GAN model [8].

We hope that the practical difficulties of training with saddle point objectives are not an underlying feature of the model but instead can be overcome with novel optimization algorithms. Further investigations, such as [30], are needed to investigate and hopefully overcome these difficulties.

*Acknowledgements.* We thank Ferenc Huszár for discussions on the generative-adversarial approach.

## Footnotes

[1] Note that for numerical implementation we recommend directly implementing the scalar function $f^*(g_f(\cdot))$ robustly instead of evaluating the two functions in sequence; see Figure 1.

[2]http://torch.ch/blog/2015/11/13/gan.html

[3]Available at `https://github.com/goodfeli/adversarial`

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
