[Supplementary Material]

# Supplementary Materials for:
# "$f$-GAN: Training Generative Neural Samplers using Variational Divergence Minimization"

**Sebastian Nowozin,  Botond Cseke,  Ryota Tomioka**
Machine Intelligence and Perception Group
Microsoft Research
{Sebastian.Nowozin, Botond.Cseke, ryoto}@microsoft.com

## 1  Introduction

We provide additional material to support the content presented in the paper. The text is structured as follows. In Section 2 we present an extended list of f-divergences, corresponding generator functions and their convex conjugates. In Section 3 we provide the proof of Theorem 1 from Section 3. In Section 5 we discuss the differences between current (to our knowledge) GAN optimisation algorithms. Section 6 provides a proof of concept of our approach by fitting a Gaussian to a mixture of Gaussians using various divergence measures. Finally, in Section 7 we present the details of the network architectures used in Section 4 of the main text.

## 2  $f$-divergences and Generator-Conjugate Pairs

In Table 2 we show an extended list of f-divergences $D_f(P\|Q)$ together with their generators $f(u)$ and the corresponding optimal variational functions $T^*(x)$. For all divergences we have $f : \mathrm{dom}_f \to \mathbb{R} \cup \{+\infty\}$, where $f$ is convex and lower-semicontinuous. Also we have $f(1) = 0$ which ensures that $D_f(P\|P) = 0$ for any distribution $P$. As shown by [1] GAN is related to the Jensen-Shannon divergence through $D_{\mathrm{GAN}} = 2D_{\mathrm{JS}} - \log(4)$. The GAN generator function $f$ does not satisfy $f(1) = 0$ hence $D_{\mathrm{GAN}}(P\|P) \neq 0$.

Table 3 lists the convex conjugate functions $f^*(t)$ of the generator functions $f(u)$ in Table 2, their domains, as well as the activation functions $g_f$ we use in the last layers of the generator networks to obtain a correct mapping of the network outputs into the domains of the conjugate functions.

The panels of Figure 1 show the generator functions and the corresponding convex conjugate functions for a variety of f-divergences.

## 3  Proof of Theorem 1

In this section we present the proof of Theorem 1 from Section 3 of the main text. For completeness, we reiterate the conditions and the theorem.

We assume that $F$ is strongly convex in $\theta$ and strongly concave in $\omega$ such that

$$\nabla_\theta F(\theta^*, \omega^*) = 0, \quad \nabla_\omega F(\theta^*, \omega^*) = 0, \tag{1}$$

$$\nabla_\theta^2 F(\theta, \omega) \succeq \delta I, \quad \nabla_\omega^2 F(\theta, \omega) \preceq -\delta I. \tag{2}$$

These assumptions are necessary except for the "strong" part in order to define the type of saddle points that are valid solutions of our variational framework.

We define $\pi^t = (\theta^t, \omega^t)$ and use the notation

$$\nabla F(\pi) = \begin{pmatrix} \nabla_\theta F(\theta, \omega) \\ \nabla_\omega F(\theta, \omega) \end{pmatrix}, \quad \tilde{\nabla} F(\pi) = \begin{pmatrix} -\nabla_\theta F(\theta, \omega) \\ \nabla_\omega F(\theta, \omega) \end{pmatrix}.$$

Figure 1: Generator-conjugate $(f, f^*)$ pairs in the variational framework of Nguyen et al. [2]. Left: generator functions $f$ used in the $f$-divergence $D_f(P\|Q) = \int_{\mathcal{X}} q(x) f\left(\frac{p(x)}{q(x)}\right) \mathrm{d}x$. Right: conjugate functions $f^*$ in the variational divergence lower bound $D_f(P\|Q) \geq \sup_{T \in \mathcal{T}} \int_{\mathcal{X}} p(x)\, T(x) - q(x) f^*(T(x))\, \mathrm{d}x$.

With this notation, Algorithm 1 in the main text can be written as

$$\pi^{t+1} = \pi^t + \eta \tilde{\nabla} F(\pi^t).$$

Given the above assumptions and notation, in Section 3 of the main text we formulate the following theorem.

**Theorem 1.** *Suppose that there is a saddle point $\pi^* = (\theta^*, \omega^*)$ with a neighborhood that satisfies conditions (1) and (2). Moreover we define $J(\pi) = \frac{1}{2}\|\nabla F(\pi)\|_2^2$ and assume that in the above neighborhood, $F$ is sufficiently smooth so that there is a constant $L > 0$ and*

$$J(\pi') \leq J(\pi) + \langle \nabla J(\pi), \pi' - \pi \rangle + \frac{L}{2}\|\pi' - \pi\|_2^2 \tag{3}$$

*for any $\pi, \pi'$ in the neighborhood of $\pi^*$. Then using the step-size $\eta = \delta/L$ in Algorithm 1, we have*

$$J(\pi^t) \leq \left(1 - \frac{\delta^2}{L}\right)^t J(\pi^0)$$

*where $L$ is the smoothness parameter of $J$. That is, the squared norm of the gradient $\nabla F(\pi)$ decreases geometrically.*

*Proof.* First, note that the gradient of $J$ can be written as

$$\nabla J(\pi) = \nabla^2 F(\pi) \nabla F(\pi).$$

Therefore we notice that,

$$
\begin{aligned}
\left\langle \tilde{\nabla} F(\pi), \nabla J(\pi) \right\rangle &= \left\langle \tilde{\nabla} F(\pi), \nabla^2 F(\pi) \nabla F(\pi) \right\rangle \\
&= \left\langle \begin{pmatrix} -\nabla_\theta F(\theta, \omega) \\ \nabla_\omega F(\theta, \omega) \end{pmatrix}, \begin{pmatrix} \nabla_\theta^2 F(\theta, \omega) & \nabla_\theta \nabla_\omega F(\theta, \omega) \\ \nabla_\omega \nabla_\theta F(\theta, \omega) & \nabla_\omega^2 F(\theta, \omega) \end{pmatrix} \begin{pmatrix} \nabla_\theta F(\theta, \omega) \\ \nabla_\omega F(\theta, \omega) \end{pmatrix} \right\rangle \\
&= -\left\langle \nabla_\theta F(\theta, \omega), \nabla_\theta^2 F(\theta, \omega) \nabla_\theta F(\theta, \omega) \right\rangle + \left\langle \nabla_\omega F(\theta, \omega), \nabla_\omega^2 F(\theta, \omega) \nabla_\omega F(\theta, \omega) \right\rangle \\
&\leq -\delta \left( \|\nabla_\theta F(\theta, \omega)\|_2^2 + \|\nabla_\omega F(\theta, \omega)\|_2^2 \right) = -\delta \|\nabla F(\pi)\|_2^2 \tag{4}
\end{aligned}
$$

In other words, Algorithm 1 decreases $J$ by an amount proportional to the squared norm of $\nabla F(\pi)$.

Now combining the smoothness (3) with Algorithm 1, we get

$$
\begin{aligned}
J(\pi^{t+1}) &\leq J(\pi^t) + \eta \left\langle \nabla J(\pi^t), \tilde{\nabla} F(\pi^t) \right\rangle + \frac{L\eta^2}{2} \|\tilde{\nabla} F(\pi^t)\|_2^2 \\
&\leq \left(1 - 2\delta\eta + L\eta^2\right) J(\pi^t) \\
&= \left(1 - \frac{\delta^2}{L}\right) J(\pi^t),
\end{aligned}
$$

where we used sufficient decrease (4) and $J(\pi) = \frac{1}{2}\|\nabla F(\pi)\|_2^2 = \frac{1}{2}\|\tilde{\nabla} F(\pi)\|_2^2$ in the second inequality, and the final equality follows by taking $\eta = \delta/L$. $\square$

| Algorithm | Maximisation in $\omega$ | Minimisation in $\theta$ |
|---|---|---|
| NCE [5] | $\alpha = 1, \beta = 0$ | NA |
| GAN-1 [1] | $\alpha = 1, \beta = 0$ | $\alpha = 1, \beta = 0$ |
| GAN-2 [1] | $\alpha = 1, \beta = 0$ | $\alpha = 0, \beta = 1$ |
| GAN-3 [4] | $\alpha = 1, \beta = 0$ | $\alpha = 1, \beta = 1$ |

Table 1: Optimisation algorithms for the GAN objective (5).

## 4 Proof of the Generalized Heuristic

Formally we prove the following statement.

**Theorem 2.** *maximizing $\mathbb{E}_{x \sim Q_\theta} \left[ g_f \left( V_\omega(x) \right) \right]$ with respect to $\theta$ has the same stationary point as minimizing $\mathbb{E}_{x \sim Q_\theta} \left[ -f^* \left( g_f(V_\omega(x)) \right) \right]$.*

*Proof.* The derivative with respect to the two objectives can be written as follows:

$$\mathbb{E}_z \left[ \frac{\mathrm{d}g_f(v)}{\mathrm{d}v} \bigg|_{v=V_\omega(G_\theta(z))} \cdot \frac{\mathrm{d}V_\omega(x)}{\mathrm{d}x} \bigg|_{x=G_\theta(z)} \cdot \frac{\partial G_\theta(z)}{\partial \theta} \right],$$

$$\mathbb{E}_z \left[ -\frac{\mathrm{d}f^*(t)}{\mathrm{d}t} \bigg|_{t=g_f(V_\omega(G_\theta(z)))} \cdot \frac{\mathrm{d}g_f(v)}{\mathrm{d}v} \bigg|_{v=V_\omega(G_\theta(z))} \cdot \frac{\mathrm{d}V_\omega(x)}{\mathrm{d}x} \bigg|_{x=G_\theta(z)} \cdot \frac{\partial G_\theta(z)}{\partial \theta} \right].$$

Thus the difference lies only in the leading term $\mathrm{d}f^*(t)/\mathrm{d}t$ evaluated at $t = g_f(V_\omega(G_\theta(z)))$. Now using (5) in the main text in the other direction, we observe that $\mathrm{d}f^*(t)/\mathrm{d}t = p(x)/q(x)$ evaluated at $t = T^*(x)$. Thus at optimality $p(x)/q(x) = 1$ and the leading term becomes a constant. $\qquad \square$

## 5 Related Algorithms

Due to recent interest in GAN type models, there have been attempts to derive various divergence measures, objective functions and algorithms. In particular, an alternative Jensen-Shannon divergence has been derived in [3] and a heuristic algorithm that behaves similarly to the one resulting from it has been proposed in [4].

In this section we summarise (some of) the current algorithms and show how they are related. Note that some algorithms use heuristics that do not correspond to a saddle point optimisation, that is, in the corresponding maximization and minimization steps they optimise alternative objectives that do not add up to a coherent joint objective. We include a short discussion of [5] because it can be viewed as a special case of GAN.

To illustrate how the discussed algorithms work, we define the objective function

$$F(\theta, \omega; \alpha, \beta) = E_{x \sim P}[\log D_\omega(x)] + \alpha E_{x \sim Q_\theta}[\log(1 - D_\omega(x))] - \beta E_{x \sim Q_\theta}[\log(D_\omega(x))], \quad (5)$$

where we introduce two scalar parameters, $\alpha$ and $\beta$, to help us highlight the differences between the algorithms shown in Table 1.

**Noise-Contrastive Estimation (NCE)**

NCE [5] is a method that estimates the parameters of an unnormalised model $p(x; \omega)$ by performing non-linear logistic regression to discriminate between the data generated from the model and some artificially generated data. To achieve this NCE casts the estimation problem as a ML estimation in a binary classification problem where the data is augmented with artificially generated data. The "true" data items are labeled as positives while the artificially generated data items are labeled as negatives. The discriminant function is defined as $D_\omega(x) = p(x; \omega)/(p(x; \omega) + q(x))$ where $q(x)$ denotes the distribution of the artificially generated data, typically a Gaussian parameterised by the empirical mean and covariance of the true data. ML estimation in this binary classification model results in an objective that has the form (5) with $\alpha = 1$ amd $\beta = 0$, where the expectations are taken w.r.t. the empirical distribution of augmented data. As a result, NCE can be viewed as a special case of

GAN where the generator is fixed and one only have maximise the objective w.r.t. the parameters of the discriminator. Another difference is that in this case the data distribution is learned through the discriminator not the generator, however, the method has many conceptual similarities to GAN.

**GAN-1 and GAN-2**

The first algorithm (GAN-1) proposed in [1] performs a stochastic gradient ascent-descent on the objective with $\alpha = 1$ and $\beta = 0$. However, the authors point out that in practice it is more advantageous to minimise $-E_{x \sim Q_\theta}[\log D_\omega(x)]$ instead of $E_{x \sim Q_\theta}[\log(1 - D_\omega(x))]$, we denote this by GAN-2. This is motivated by the observation that in the early stages of training when $Q_\theta$ is not sufficiently well fitted, $D_\omega$ can saturate fast leading to weak gradients in $E_{x \sim Q_\theta}[\log(1 - D_\omega(x))]$. The $-E_{x \sim Q_\theta}[\log D_\omega(x)]$ term, however, can provide stronger gradients and leads to the same fixed point. This heuristic can be viewed as using $\alpha = 1, \beta = 0$ in the maximisation step and $\alpha = 0, \beta = 1$ in the minimisation step[1].

**GAN-3**

In [4] a further heuristic for the minimisation step is proposed. Formally, it can be viewed as a combination of the minimisation steps in GAN-1 and GAN-2. In the proposed algorithm, the maximisation step is performed similarly ($\alpha = 1, \beta = 0$), but the minimisation is done using $\alpha = 1$ and $\beta = 1$. This choice is motivated by KL optimality arguments. The author argues that since the optimal discriminator is given by

$$D^*(x) = \frac{p(x)}{q_\theta(x) + p(x)} \tag{6}$$

when close to optimality, the minimisation of $E_{x \sim Q_\theta}[\log(1 - D_\omega(x))] - E_{x \sim Q_\theta}[\log D_\omega(x)]$ corresponds to the minimisation of the reverse KL divergence $E_{x \sim Q_\theta}[\log(q_\theta(x)/p(x))]$. This approach can be viewed as choosing $\alpha = 1$ and $\beta = 1$ in the minimisation step.

**Remarks on the Weighted Jensen-Shannon Divergence in [3]**

The GAN/variational objective corresponding to alternative Jensen-Shannon divergence measure proposed in [3] (see Jensen-Shannon-weighted in Table 1) is

$$F(\theta, \omega; \pi) = E_{x \sim P}[\log D_\omega(x)] - (1 - \pi)E_{x \sim Q_\theta}\left[\log \frac{1 - \pi}{1 - \pi D_\omega(x)^{1/\pi}}\right]. \tag{7}$$

Note that we have the $T_\omega(x) = \log D_\omega(x)$ correspondence. According to the definition of the variational objective, when $T_\omega$ is close to optimal then in the minimisation step the objective function is close to the chosen divergence. In this case the optimal discriminator is

$$D^*(x)^{1/\pi} = \frac{p(x)}{(1 - \pi)q_\theta(x) + \pi p(x)}. \tag{8}$$

The objective in (7) vanishes when $\pi \in \{0, 1\}$, however, when $\pi$ is only is close to $0$ and $1$, it can behave similarly to the KL and reverse KL objectives, respectively. Overall, the connection between GAN-3 and the optimisation of (7) can only be considered as approximate. To obtain an exact KL or reverse KL behavior one can use the corresponding $f$-GAN objectives. For a simple illustration of how the $f$-divergences and $f$-GAN objectives behave see Section 2.5 and Section 6 below.

# 6 Details of the Univariate Example

We follow up on the example in Section 2.5 of the main text by presenting further details about the quality and behavior of the approximations resulting from using various $f$-divergence measures. For completeness, we reiterate the setup and then we present further results.

Figure 2: Gaussian approximation of a mixture of Gaussians. Gaussian approximations obtained by direct optimisation of $D_f(p||q_{\theta*})$ (dashed-black) and the optimisation of $F(\hat{\omega}, \hat{\theta})$ (solid-colored). Right-bottom: optimal variational functions $T^*$ (dashed) and $T_{\hat{\omega}}$ (solid-red).

**Setup.** We approximate a mixture of Gaussian [2] by learning a Gaussian distribution. The model $Q_\theta$ is represented by a linear function which receives a random noise $z \sim \mathcal{N}(0,1)$ and outputs

$$G_\theta(z) = \mu + \sigma z, \tag{9}$$

where $\theta = (\mu, \sigma)$ are the parameters to be learned. For the variational function $T_\omega$ we use the neural network

$$x \quad \to \text{Linear}(1,64) \to \text{Tanh} \to \text{Linear}(64,64) \to \text{Tanh} \to \text{Linear}(64,1). \tag{10}$$

We optimise the objective $F(\omega, \theta)$ by using the single-step gradient method presented in Section 3.1 of the main text . In each step we sample batches of size $1024$ from $p(x)$ and $p(z)$ and we use a step-size of $0.01$ for updating both $\omega$ and $\theta$. We compare the results to the best fit provided by the exact optimisation of $D_f(P\|Q_\theta)$ w.r.t. $\theta$, which is feasible in this case by solving the required integrals numerically. We use $(\hat{\omega}, \hat{\theta})$ (learned) and $\theta^*$ (best fit) to distinguish the parameters sets used in these two approaches.

**Results.** The panels in Figure 2 shows the density function of the data distribution as well as the Gaussian approximations corresponding to a few $f$-divergences form Table 2. As expected, the KL approximation covers the data distribution by fitting its mean and variance while KL-rev has more of a mode-seeking behavior [6]. The fit corresponding to the Jensen-Shannon divergence is somewhere between KL and KL-rev. All Gaussian approximations resulting from neural network training are close to the ones obtained by direct optimisation of the divergence (learned vs. best fit).

In the right–bottom panel of Figure 2 we compare the variational functions $T_{\hat{\omega}}$ and $T^*$. The latter is defined as $T^*(x) = f'(p(x)/q_{\theta^*}(x))$, see main text. The objective value corresponding to $T^*$ is the true divergence $D_f(P\|Q_{\theta^*})$. In the majority of the cases our $T_{\hat{\omega}}$ is close to $T^*$ in the area of interest. The discrepancies around the tails can be due to (1) the class of functions resulting from the $\tanh$ activation function has limited capability representing the tails, and (2) in the Gaussian case there is a lack of data in the tails. These limitations, however, do not have a significant effect on the learned parameters.

# 7 Details of the Experiments

In this section we present the technical setup as well as the architectures we used in the experiments described in Section 4.

## 7.1 Deep Learning Environment

We use the deep learning framework *Chainer* [7], version 1.8.1, running on CUDA 7.5 with CuDNN v5 on NVIDIA GTX TITAN X.

## 7.2 MNIST Setup

**MNIST Generator**

$$z \quad \to \text{Linear}(100, 1200) \to \text{BN} \to \text{ReLU} \to \text{Linear}(1200, 1200) \to \text{BN} \to \text{ReLU}$$
$$\to \text{Linear}(1200, 784) \to \text{Sigmoid} \tag{11}$$

All weights are initialized at a weight scale of $0.05$, as in [1].

**MNIST Variational Function**

$$x \quad \to \text{Linear}(784,240) \to \text{ELU} \to \text{Linear}(240,240) \to \text{ELU} \to \text{Linear}(240,1), \tag{12}$$

where ELU is the exponential linear unit [8]. All weights are initialized at a weight scale of $0.005$, one order of magnitude smaller than in [1].

**Variational Autoencoders** For the variational autoencoders [9], we used the example implementation included with *Chainer* [7]. We trained for 100 epochs with 20 latent dimensions.

## 7.3 LSUN Natural Images

In the LSUN experiment we use the generator

$$
\begin{aligned}
z \quad &\to \text{Linear}(100, 6 \cdot 6 \cdot 512) \to \text{BN} \to \text{ReLU} \to \text{Reshape}(512,6,6) \\
&\to \text{Deconv}(512,256) \to \text{BN} \to \text{ReLU} \to \text{Deconv}(256,128) \to \text{BN} \to \text{ReLU} \\
&\to \text{Deconv}(128,64) \to \text{BN} \to \text{ReLU} \to \text{Deconv}(64,3), \quad\quad\quad\quad (13)
\end{aligned}
$$

where all Deconv operations use a kernel size of four and a stride of two.

Table 2: List of $f$-divergences $D_f(P\|Q)$, their generator functions and the optimal variational functions.

| Name | $D_f(P\|Q)$ | Generator $f(u)$ | $T^*(x)$ |
|---|---|---|---|
| Total variation | $\frac{1}{2}\int \lvert p(x) - q(x)\rvert\,\mathrm{d}x$ | $\frac{1}{2}\lvert u - 1\rvert$ | $\frac{1}{2}\operatorname{sign}\!\left(\frac{p(x)}{q(x)} - 1\right)$ |
| Kullback-Leibler | $\int p(x)\log\frac{p(x)}{q(x)}\,\mathrm{d}x$ | $u\log u$ | $1 + \log\frac{p(x)}{q(x)}$ |
| Reverse Kullback-Leibler | $\int q(x)\log\frac{q(x)}{p(x)}\,\mathrm{d}x$ | $-\log u$ | $-\frac{q(x)}{p(x)}$ |
| Pearson $\chi^2$ | $\int \frac{(q(x)-p(x))^2}{p(x)}\,\mathrm{d}x$ | $(u-1)^2$ | $2\left(\frac{p(x)}{q(x)} - 1\right)$ |
| Neyman $\chi^2$ | $\int \frac{(p(x)-q(x))^2}{q(x)}\,\mathrm{d}x$ | $\frac{(1-u)^2}{u}$ | $1 - \left[\frac{q(x)}{p(x)}\right]^2$ |
| Squared Hellinger | $\int \left(\sqrt{p(x)} - \sqrt{q(x)}\right)^2\,\mathrm{d}x$ | $\left(\sqrt{u}-1\right)^2$ | $\left(\sqrt{\frac{p(x)}{q(x)}} - 1\right)\cdot\sqrt{\frac{q(x)}{p(x)}}$ |
| Jeffrey | $\int (p(x)-q(x))\log\left(\frac{p(x)}{q(x)}\right)\,\mathrm{d}x$ | $(u-1)\log u$ | $1 + \log\frac{p(x)}{q(x)} - \frac{q(x)}{p(x)}$ |
| Jensen-Shannon | $\frac{1}{2}\int p(x)\log\frac{2p(x)}{p(x)+q(x)} + q(x)\log\frac{2q(x)}{p(x)+q(x)}\,\mathrm{d}x$ | $-(u+1)\log\frac{1+u}{2} + u\log u$ | $\log\frac{2p(x)}{p(x)+q(x)}$ |
| Jensen-Shannon-weighted | $\pi\int p(x)\log\frac{p(x)}{\pi p(x)+(1-\pi)q(x)} + (1-\pi)q(x)\log\frac{q(x)}{\pi p(x)+(1-\pi)q(x)}\,\mathrm{d}x$ | $\pi u\log u - (1-\pi+\pi u)\log(1-\pi+\pi u)$ | $\pi\log\frac{p(x)}{(1-\pi)q(x)+\pi p(x)}$ |
| GAN | $\int p(x)\log\frac{2p(x)}{p(x)+q(x)} + q(x)\log\frac{2q(x)}{p(x)+q(x)}\,\mathrm{d}x - \log(4)$ | $u\log u - (u+1)\log(u+1)$ | $\log\frac{p(x)}{p(x)+q(x)}$ |
| $\alpha$-divergence ($\alpha \notin \{0,1\}$) | $\frac{1}{\alpha(\alpha-1)}\int \left(p(x)\left[\left(\frac{q(x)}{p(x)}\right)^\alpha - 1\right] - \alpha(q(x) - p(x))\right)\,\mathrm{d}x$ | $\frac{1}{\alpha(\alpha-1)}\left(u^\alpha - 1 - \alpha(u-1)\right)$ | $\frac{1}{\alpha-1}\left(\left[\frac{p(x)}{q(x)}\right]^{\alpha-1} - 1\right)$ |

| Name | Output activation $g_f$ | $\mathrm{dom}_{f^*}$ | Conjugate $f^*(t)$ | $f'(1)$ |
|---|---|---|---|---|
| Total variation | $\frac{1}{2}\tanh(v)$ | $-\frac{1}{2}\le t\le\frac{1}{2}$ | $t$ | 0 |
| Kullback-Leibler (KL) | $v$ | $\mathbb{R}$ | $\exp(t-1)$ | 1 |
| Reverse KL | $-\exp(v)$ | $\mathbb{R}_-$ | $-1-\log(-t)$ | $-1$ |
| Pearson $\chi^2$ | $v$ | $\mathbb{R}$ | $\frac{1}{4}t^2+t$ | 0 |
| Neyman $\chi^2$ | $1-\exp(v)$ | $t<1$ | $2-2\sqrt{1-t}$ | 0 |
| Squared Hellinger | $1-\exp(v)$ | $t<1$ | $\frac{t}{1-t}$ | 0 |
| Jeffrey | $v$ | $\mathbb{R}$ | $W(e^{1-t})+\frac{1}{W(e^{1-t})}+t-2$ | 0 |
| Jensen-Shannon | $\log(2)-\log(1+\exp(-v))$ | $t<\log(2)$ | $-\log(2-\exp(t))$ | 0 |
| Jensen-Shannon-weighted | $-\pi\log\pi-\log(1+\exp(-v))$ | $t<-\pi\log\pi$ | $(1-\pi)\log\frac{1-\pi}{1-\pi e^{t/\pi}}$ | 0 |
| GAN | $-\log(1+\exp(-v))$ | $\mathbb{R}_-$ | $-\log(1-\exp(t))$ | $-\log(2)$ |
| $\alpha$-div. ($\alpha<1,\alpha\neq0$) | $\frac{1}{1-\alpha}-\log(1+\exp(-v))$ | $t<\frac{1}{1-\alpha}$ | $\frac{1}{\alpha}(t(\alpha-1)+1)^{\frac{\alpha}{\alpha-1}}-\frac{1}{\alpha}$ | 0 |
| $\alpha$-div. ($\alpha>1$) | $v$ | $\mathbb{R}$ | $\frac{1}{\alpha}(t(\alpha-1)+1)^{\frac{\alpha}{\alpha-1}}-\frac{1}{\alpha}$ | 0 |

Table 3: Recommended final layer activation functions and critical variational function level defined by $f'(1)$. The objective function for training a generative neural network $G_\theta$ given a true distribution $P$ and an auxiliary variational function $T$ is $\min_\theta \max_T (\mathbb{E}_{x\sim P}[T(x)] - \mathbb{E}_{x\sim G_\theta}[f^*(T(x))])$. For any sample $x$ the variational function produces a scalar $v(x)\in\mathbb{R}$. The output activation provides a differentiable map $g:\mathbb{R}\to\mathrm{dom}_{f^*}$, defining $T(x)=g(v(x))$. The critical value $f'(1)$ can be interpreted as a classification threshold applied to $T(x)$ to distinguish between true and generated samples. $W$ is the Lambert-$W$ product log function.

## Footnotes

[1] A somewhat similar observation regarding the artificially generated data is made in [5]: in order to have meaningful training one should choose the artificially generated data to be close the the true data, hence the choice of an ML multivariate Gaussian.

[2]The plots on Figure 2 correspond to $p(x) = (1-w)N(x; m_1, v_1) + wN(x; m_2, v_2)$ with $w = 0.67, m_1 = -1, v_1 = 0.0625, m_2 = 2, v_2 = 2$.