[Reviews · NeurIPS 2016]

Reviewer 1

Summary

The authors provide an interesting extension to GANs by showing that any f-divergence can be used as the objective rather than the original JSD. However, the results seem weak. They show inconclusive results on MNIST and the qualitative results on LSUN for each divergence look similar.

Qualitative Assessment

Nicely written and readable paper. The construction of the VDM approach is elegant and is an interesting extension of Nguyen et al. from estimating a divergence to estimating model parameters. The results also show that there are no issues optimizing the other f-divergences and it provides a reason why Goodfellow et al.'s modified objective works better. It would also be interesting to know if the other f-divergences or Alg. 1 make GAN training more stable. The experiments on MNIST are unconvincing as they stand. I also wonder if the KDE estimator approach will be biased towards a particular divergence. It would be nice to include human evaluation for a task such as LSUN to provide a more quantitative result or show the effect of the divergences on limited training data. Also, in line 220 by 'not rich enough' do you mean has not enough capacity? Minor comments/typos: Line 29: model -> models Line 35: networks -> network Line 36: JS is a different font to KL here. Line 37: Long, hard to read sentence. Line 82: Bad sentence. Line 111: 'is' is misplaced. Line 130: By 'again' do you mean with a different divergence? Line 250: 'the' typo Line 251: 'funcationand' typo Line 260: 'similar', insert to. Reference 15,16,25 should be to ICLR not arXiv.

Confidence in this Review

2-Confident (read it all; understood it all reasonably well)


Reviewer 2

Summary

This paper introduces Variational Divergence Minimization (VDM), a novel training criteria for generative models based on f-divergence minimization. Starting from the idea of learning a model Q by minimizing the f-divergence between the empirical distribution P and model Q, the authors derive a mini-max objective function which generalizes the objective optimized by Generative Adversarial Networks (GAN). The GAN objective is recovered for a particular choice of generator function f(u), linked to the Jensen-Shannon divergence. Interestingly, other choices of f(u) can lead to GAN-like objectives which minimize KL-divergence (in either direction) or other f-divergences. Experiments confirm the efficacy of the generalized GAN objective, with visualizations of samples and coarse likelihood estimates (via non-parametric density estimate) on a synthetic toy dataset, MNIST and LSUN datasets.

Qualitative Assessment

Technical quality I am currently on the fence with respect to technical quality, but hope the authors can clarify the following in the rebuttal. The starting point for the method is a divergence D_f(P||Q) which we aim to minimize. Unfortunately, the mini-max objective function of Eq. (6) is a lower-bound on this divergence. This seems problematic as optimizing Eq (6) would then not guarantee anything with respect to the original divergence, regardless of how tight the bound is. This is in stark contrast to variational EM, which maximizes a lower-bound on the log-likelihood, a quantity we also aim to maximize. Another issue with the paper is the evaluation of trained models. As log-likelihoods cannot be computed, the evaluation of trained models is limited to (1) visualization of samples and (2) a kernel density estimation technique. Given the issues with the latter, I would recommend the authors include other evaluation metrics such as inpainting, or class-conditional image generation as in the recent “Conditional Image Generation with PixelCNN Decoders” paper. Potential impact or usefulness I expect reviews to diverge on this question. On one hand, this is mostly a theoretical paper with perhaps limited appeal because of it. While the generalization of GANs to the broader class of f-divergences is interesting from a theoretical point of view, the experimental section does not convincingly show the usefulness of this generalization. Samples obtained via different choices of generator function seem comparable, while the likelihood evaluation on MNIST is hampered by the reliance on the non-parametric density estimator. On the other hand, this paper brings much needed theory to the field of GANs which have been gaining momentum in the community. It could also potentially bridge the gap between GANs and maximum likelihood based approaches like VAEs. It is particularly interesting that minimization of KL(P||Q) or KL(Q||P) are both special cases of the proposed VDM framework, as it has been argued that the mode-seeking behavior of the latter may be preferable for generative models. To this end, I find it regrettable that the authors did not provide results using the reverse KL generator function on the LSUN dataset. Novelty/originality While the connection between GANs and Jensen-Shannon divergence was already known, this paper generalizes the objective to the wider class of f-divergences. I believe this is a significantly novel contribution. Unfortunately, the “single step gradient method” proposed in the paper is not as novel as the authors claim. While the original paper did advocate performing k-steps of optimization on the discriminator, before adapting the generator network, the authors actually used k=1 in practice. The resulting algorithm remain different in that setting k=1 performs block-coordinate descent (first adapt classifier, then adapt generator for fixed classifier) while the authors proposed to optimize both sets of parameters jointly. This point ought to be clarified in the paper. Clarity and presentation The paper is generally well written and (thankfully) provides the necessary background material on f-divergences.

Confidence in this Review

2-Confident (read it all; understood it all reasonably well)


Reviewer 3

Summary

This paper shows how the GAN framework can be extended to train the model with many different divergences, rather than just the Jensen-Shannon Divergence.

Qualitative Assessment

I think this paper should be accepted. The authors should feel free to take a few sentences from what I say here and put it in the paper: Currently, many researchers working on generative models are interested in the differences between GANs and VAEs. A general consensus, prior to the appearance of this paper on ArXiv, was that VAEs were good at assigning high likelihood to data points, and GANs were good at generating realistic samples, and that this difference is due to the differing cost functions used. Because this paper shows that GANs behave similarly whether they are trained with the KL divergence cost or the cost proposed in [10], that changes the interpretation of the situation. It now seems that GAN samples are high quality due to something about their training procedure or the way the model family is specified, not because of the cost function used to train them. So this paper provides important evidence that is relevant to settling an important scientific question. It's also possible that the specific divergences provided here might find practical application. Here are a few things that should be improved, but that I don't think are dealbreakers: - There's one big technical problem here, which is that the bound goes the wrong way. Usually for variational methods, we want to minimize an upper bound on the cost, but here we are minimizing a lower bound. That means that in the general case, the algorithm may exhibit strong non-convergence behavior. This problem is discussed a little bit in this ICLR workshop paper: https://arxiv.org/pdf/1412.6515v4.pdf We also don't really know how much we are improving the true cost, because we don't have any idea how tight the bound is. The algorithm seems to be at least somewhat sane in practice, so it isn't necessarily broken, but it's worth discussing in the paper that the bound doesn't automatically give us all of the theoretical guarantees it gives in the usual variational learning setting. - The same ICLR workshop paper ( https://arxiv.org/pdf/1412.6515v4.pdf ) also shows how to modify GANs to minimize the KL divergence, but in a different way than this paper. It's worth discussing this in the related work. This work shows how to construct a bound on the KL. The workshop paper shows how to make a stochastic approximation to the gradient of the KL that would be unbiased if the discriminator were optimal, but in practice is subject to overfitting and underfitting in the discriminator. The technique in the workshop paper requires departing from the minimax game formulation and instead uses an independent cost for each player. - The original GAN paper [9] was accompanied by an open source code release that included the algorithm presented in sec 3.1, and several follow-up projects based on [9] have used that same procedure, so the algorithm itself is not an original contribution of this paper. However, the proof that it converges is. - Likewise, the practice of monitoring true positive rates / true negative rates is attributed to Larsen and Sønderby, but this was actually done in the original code release accompanying [9]. - Typo on line 35: "networks" instead of "network" - The review of f-divergences might explain whether there are any desirable properties of the family of f-divergences. This could help motivate the paper a bit more. As the reader, I have the general idea that for an underpowered model, some divergences result in choosing a subset of modes to capture, while others result in averaging modes together, but I don't know exactly what the benefit of having a wide family of divergences is. - Eq 4: I didn't understand why it is necessary to use Jensen's inequality to pull the sup out of the integral. In the case of discrete x, the sup could be pulled outside of the sum with strict equality, as long as the family of functions for T(x) is unrestricted. Is the use of Jensen's inequality just necessary because issues that are specific to continuous x, like restrictions to function families that are integrable / handling cases where the supremum is not actually attained so it's not possible for T(x) to take on a value that results in the supremum? (This is a minor point because I agree that the restriction of the family of T(x) requires an inequality) - In the discussion of Eq 4, it's worth mentioning that in practice, part of the reason we'll have an inequality is that T(x) will be imperfectly optimized. - Numerous issues with Table 2: First, there's one issue with the evaluation itself: The table on the right seems to show that the variational bound reaches nearly its optimal value, but because we don't see (estimates of) the true divergence, we don't have any idea how tight that bound is. The rest of the issues with table 2 are related to clarity. I spent approximately half an hour reading it, feeling like I was taking an IQ test. These should be easy to fix, given a specification of what confused me: 1) I don't understand what "objective values to the true distribution" means. 2) I wasn't sure what the F values in the table in the left were. Are these: a) The theoretical minimum of F b) The value of F found by your training algorithm c) The value of F attained when D is at its theoretical minimal? I think the answer is (a), based on issue #4 3) Are the values in the table on the right the lower bounds or the actual divergence values? I think they are the lower bounds because some of them go lower than the optimal D_f value shown in the table on the left. I would suggest clearly indicating them as bounds by calling them \hat{KL} rather than KL, \hat{JS} rather than JS, etc. 4) In the table on the right, some values are lower than their minimum value reported on the left. For example, training and testing with KL-rev yields .2414, but in the table on the left its optimal value is reported as .2415. The same thing also happens with JS, with a larger discrepancy. Initially I thought this was an error in the table, but then I realized it is possible I misinterpreted what was being reported for F in the table on the left. Full disclosure: I've seen this paper on ArXiv, before I was assigned to review it. I didn't know the authors previously, and don't think that knowledge of who wrote it has influenced my ratings.

Confidence in this Review

3-Expert (read the paper in detail, know the area, quite certain of my opinion)


Reviewer 4

Summary

The author generalizes the generative adversarial network objective to a rich family of divergences and explore their different behaviors. This includes kullback-leibler and reverse kullback leibler divergences. They also provide a simplified version of the optimization algorithm. Experiments are run on a small synthetic dataset for in depth analysis and on MNIST and LSUN for real world results.

Qualitative Assessment

The paper was very well written. The theoretical side was non trivial and the experiments were complete, honest and contained enough information for reproducibility. Congrats for the great paper. Typos: * Line 82: plot of the ? * Line 139: the choice of taking? * Line 191: report the number of epochs instead of number of hours. * Table 4: Define SEM * Line 207: as as * Line 250: the there

Confidence in this Review

2-Confident (read it all; understood it all reasonably well)


Reviewer 5

Summary

It is well known that GAN actually aims to minimize JSD divergence, i.e., JSD(p_x || q_x). This paper tries to generalize JSD to more general divergence definition (f-divergence). By using Fenchel Conjugate dual, GAN is formulated as a minmax problem, i.e., minimizing variational low bound of f-divergence. Though, Ian Goodfellow has done similar analyses/claims in [9] and "On Distinguishability Criteria for Estimating Generative Models", such variational framework is not clear and well formulated. GAN aims to do alternative gradient (outer loop: \min, inner loop: \max), this paper proposes a single-step gradient method which tries to update both w and \theta in a single back-propagation.

Qualitative Assessment

Strength: [1] Using Fenchel Conjugate trick to explain minmax formulation in GAN (Eq.(4)) is novel and quite nice. [2] Generalization to f-divergence and comparison of different divergence is quite useful Weakness: [1] Some claims are not well explained. Experiment section should be further strengthened. Comments: [1] single-step gradient method shows gradients decrease geometrically under some conditions. However, such conditions might not be satisfied in practice. It would be nice to show experiments to compare with original optimization method in GAN in terms of speed and objective value. [2] Eqn.5 provides the best T* in theoretical analysis, but neural network architecture, e.g., activation function, is not designed explicitly according to theoretical analysis. It would be nice to show how tight the lower bound is in experiments. [3] L165-166 make sense, but what extra points authors want to claim? I can not find the proof of the statement in L167. [4] How to get threshold f'(1) in L173? More details should be nice. [5] In experiment section, it looks like some conclusions are not well supported/explained, e.g., the intuition of why some divergence performs better. Just some qualitative images are shown in Fig.3, less take away messages in LSUN Natural Images section.

Confidence in this Review

2-Confident (read it all; understood it all reasonably well)


Reviewer 6

Summary

This paper extends the variational method to estimate f-divergences for a fixed model to estimating model parameters. The authors show that this variational divergence estimation framework can be used to derive the training objective in Generative-adversarial neural networks and generalize it to any f-divergence. The paper also proposes an algorithm to minimize the variational divergence training objective and analyze its convergence. Training results on MNIST and LSUN are shown and discussed.

Qualitative Assessment

The paper proposes a nice framework to generalize the training objective in the Generative-Adversarial Neural Networks (GAN) to arbitrary f-divergence. The same framework can be used to analyze the effect of difference f-divergence on the performance of GAN. However, the paper does not show clear improvements over existing methods using the proposed framework. More specifically, it would be interesting to see classification results on benchmarks using different f-divergences. The paper is well-written. The claims, explanation, and derivation in the paper are clear and easy to follow.

Confidence in this Review

1-Less confident (might not have understood significant parts)